# Effect of Visually Induced Motion Sickness from Head-Mounted Display on Cardiac Activity

**DOI:** 10.3390/s22166213

**Published:** 2022-08-18

**Authors:** Sangin Park, Jihyeon Ha, Laehyun Kim

**Affiliations:** 1Industry-Academy Cooperation Team, Hanyang University, Seoul 04763, Korea; 2Center for Bionics, Korea Institute of Science and Technology, 5 Hwarang-ro 14-gil, Seongbuk-gu, Seoul 04763, Korea; 3Department of Biomedical Engineering, Hanyang University, Seoul 04763, Korea; 4Department of HY-KIST Bio-Convergence, Hanyang University, Seoul 04763, Korea

**Keywords:** visually induced motion sickness, normalized heart rate variability, cardiac activity, head-mounted display, cognitive load

## Abstract

Head-mounted display (HMD) virtual reality devices can facilitate positive experiences such as co-presence and deep immersion; however, motion sickness (MS) due to these experiences hinders the development of the VR industry. This paper proposes a method for assessing MS caused by watching VR content on an HMD using cardiac features. Twenty-eight undergraduate volunteers participated in the experiment by watching VR content on a 2D screen and HMD for 12 min each, and their electrocardiogram signals were measured. Cardiac features were statistically analyzed using analysis of covariance (ANCOVA). The proposed model for classifying MS was implemented in various classifiers using significant cardiac features. The results of ANCOVA reveal a significant difference between 2D and VR viewing conditions, and the correlation coefficients between the subjective ratings and cardiac features have significant results in the range of −0.377 to −0.711 (for SDNN, pNN50, and *ln* HF) and 0.653 to 0.677 (for *ln* VLF and *ln* VLF/*ln* HF ratio). Among the MS classification models, the linear support vector machine achieves the highest average accuracy of 91.1% (10-fold cross validation) and has a significant permutation test outcome. The proposed method can contribute to quantifying MS and establishing viewer-friendly VR by determining its qualities.

## 1. Introduction

Virtual reality (VR) using head-mounted displays (HMDs) has become increasingly popular for professional and entertainment purposes and contributed to technological advancement and increased economic activity [1,2]. VR technology has been used in many areas, such as military training simulations [3], training or education in medical procedures [4], architecture [5], manufacturing [6], entertainment [7], and gaming [8]. VR technology can provide an experience that is impossible in the real world and deep immersion [9]. However, the devices trigger motion sickness (MS), including visual fatigue, nausea, anxiety, and disorientation, in some users [10]. Visually induced motion sickness (VIMS), motion sickness disorder (MSD), and VIMS disorder are defined as vestibular disorders; however, MS can be experienced by anyone [11]. Approximately 33% of the population is highly susceptible [12], and at least 59% of the population has reported experiencing MS [13]. Many developers have attempted to improve software and hardware; however, the issue of MS remains [14]. Consequently, MS is a major obstacle in the popularization and development of the VR industry [15]. Thus, studies on minimizing MS are necessary, which can contribute to improving the VR user experience and friendliness. To solve the problem of MS, reliable measurement methods for quantitatively assessing MS should be established [2,16].

Many previous studies have attempted to understand and interpret MS from the perspectives of postural instability [17]; vestibular function [18]; eye movement [19]; the autonomic nervous system, such as cardiac activity, electrodermal activity, and skin temperature [1,20]; the central nervous system, such as electroencephalogram (EEG) oscillations [21] and functional magnetic resonance imaging [22]; and pupillary rhythms [23]. However, when using machine learning, most studies do not distinguish MS patients from healthy controls. Some studies have attempted to classify the level of MS severity, as shown in Table 1, using the following criteria: EEG oscillations [24,25,26,27,28,29,30,31], heart–brain connectivity [2], vision technology [16], and multimodal data fusion [32,33,34,35,36,37]. Most studies attempting to classify MS are based on brain activity and multimodal data fusion. These approaches have significant disadvantages, such as the measurement burden of sensor attachment and low usability compared to cardiac activity. Despite these limitations, the method developed in a previous study has the advantage of acquiring significant and important data. Thus, fields that require precise measurement of phenomena related MS can be at an advantage. However, a simple method should be developed to assess the sensitivity of MS, because there are fields that require rapid and easy MS measurement or monitoring. Thus, we propose developing a method that enables the simple and convenient assessment of MS based on a single measurement of cardiac activity.

Physiologically, heart rate variability (HRV) indicates the interaction of autonomic, intrinsic, and humoral influences on heart rate [38]. The HRV spectrum for assessing autonomic balance is divided into very low frequency (VLF, 0.0033–0.04 Hz), low frequency (LF, 0.04–0.15 Hz), and high frequency (HF, 0.15–0.4 Hz), which correspond to sympathetic activity, a mixture of sympathetic and parasympathetic activities, and parasympathetic activity, respectively [39,40]. Previous studies have reported that the development of MS is strongly correlated with sympathetic and parasympathetic activities [41,42]. In addition, the time domain indices of cardiac activity, such as heart rate, the standard deviation of normal to normal (SDNN), and the proportion of successive RR intervals (pNN50), have been reported to be associated with MS [43,44]. In multimodal data fusion (see Table 1), a few studies have considered the heart response intended to be used in this study to measure the MS symptom. These studies used heart rate [32,34,35,37], R-peak amplitude [32], SDNN [35], and HRV index (i.e., HRV amplitude, LF, HF and LF/HF ratio) [34] as indicators of cardiac activity. This study intends to consider a new indicator, normalized HRV, which includes cardiac features used in previous studies.

To this end, this study proposes a novel method for classifying MS caused by VR content from HMDs based on cardiac features. The cardiac features measured in both before and after VR viewing conditions are compared with 2D conditions using a statistical analysis method called analysis of covariance (ANCOVA); moreover, these features are analyzed using a partial correlation with a simulator sickness questionnaire (SSQ). Herein, MS (VR condition) is distinguished from the normal state (2D condition) using statistically significant cardiac features based on various classifiers, such as linear discriminant analysis (LDA), K-nearest neighbors (KNN), decision tree (DT), and linear support vector machine (LSVM). In addition, a real-time system is developed to monitor MS.

## 2. Methods

### 2.1. Participants

Thirty participants volunteered to participate in this study. Two participants were excluded from the experiment due to premature termination caused by severe levels of VIMS, resulting in a final sample of 28 participants (14 females and 14 males). Participants’ ages ranged from 21 to 34 years (Mean = 26.9 years, Standard deviation = 3.5 years). All participants had normal or corrected-to-normal vision and no family or medical history of cardiovascular disease. Informed written consent was obtained from each subject prior to the experiment, and they were instructed to get a full night’s rest and to abstain from cigarettes, alcohol, and caffeine for 24 h prior to the experiment. This research complied with the tenets of the Declaration of Helsinki and was approved by the Institutional Review Board of Sangmyung University (No. BE2018-46).

### 2.2. Experimental Stimuli and Procedure

VR content “Ultimate Booster Experience” (GexagonVR, Saint Petersburg, Russia, 2016) was used in this experiment to cause VIMS through the Oculus Rift S VR HMD device (Oculus VR Inc., Menlo Park, CA, USA). This study aimed to choose VR stimuli that can sufficiently cause the MS state, and the VR content “Ultimate Booster Experience” with the highest SSQ score among the five VR contents was chosen through a pre-test of 20 subjects. None of the participants had previously experienced viewing VR content (i.e., Ultimate Booster Experience) by using an HMD. The VR stimulus in the experiment contained content including a giant swing, bungee jump, air balloon, eagle flight, and rocket mode.

This study employed a “within-subject” design to compare the viewer’s cardiac activity in response to the VR contents under the 2D (non-VIMS) and VR conditions (VIMS). The participants sat on a comfortable armchair in an electrically shielded room. They were instructed to experience the VR content in 2D (viewing distance of 1 m) and VR versions using a 27-inch LED monitor (LG Electronics Inc., Seoul, Korea) and the HMD device. The scenes in the two versions were identical. Participants experienced the VR content using either the 2D or the HMD version of the VR content for 12 min. The experiment was conducted simultaneously for two days. The order of tasks (i.e., 2D and VR versions) was count-balanced randomly to minimize sequence/order effects by repeated measures design (that is, VR and 2D versions on the first and second days, respectively, or vice versa). A resting-state was provided for 5 min before and after viewing tasks, and electrocardiogram (ECG) signals were measured. The ECG signals were measured using the lead-1 method using three electrodes placed on the left collarbone (ground, black lead), left (VIN+, red lead), and right arms (VIN-, white lead), see Figure 1B. To obtain optimal electrode response, the surface of the skin where the electrode was to be placed was abraded with a clean, dry cloth. The ECG wireless sensor (ECG BioNomadix) to measure and transmit signals was placed on the stomach using a band. The experiment was conducted in an electrically shielded room to minimize the risk of external interference, which can affect ECG measurements. Except for the necessary user interface, all equipment was located outside the room using a connection cable. In addition, the participants were asked to self-report (a four-point scale, 0 to 3) subjective MS using an SSQ [45] before and after the viewing tasks. Subsequently, cardiac activity and subjective ratings before and after the viewing tasks were compared. The experimental procedure and environment are illustrated in Figure 1.

The SSQ employed is widely used to describe and assess users’ levels of MS symptoms and useful in VR studies. This SSQ comprises 16 items related to the symptoms of MS and is categorized into three non-mutually exclusive factors: (1) nausea (*N*), which comprises general discomfort, increased salivation, sweating, nausea, difficulty concentrating, stomach awareness, and burping; (2) oculomotor responses (*O*), which include general discomfort, fatigue, headache, eyestrain, difficulty focusing, difficulty concentrating, and blurred vision; and (3) disorientation (*D*), which comprises difficulty focusing, nausea, fullness of the head, blurred vision, dizziness (eyes open), dizziness (eyes closed), and vertigo. The total SSQ score is calculated using Equation (1) based on the three factors, where the values of *N*, *O*, and *D* are defined by summing the subjective rating values (4-point scale, 0–3) of each questionnaire for nausea, oculomotor responses, and disorientation, respectively. In general, a five-point scale is used for the single stimulus subjective tests as recommended in ITU-T REC. P.913 [46], but this work performed a subjective rating using the 4-point scale (0, 1, 2, 3) according to guidelines in a previous study [45]. Examples of SSQ are shown in Appendix A.
$$\mathrm{Total}\;\mathrm{SSQ}\;\mathrm{score}=\left(N+O+D\right)\times 3.74$$
(1)N score=N×9.54, O score = O ×7.58, D score=D×13.92

### 2.3. Data Acquisition and Signal Processing

ECG signals were recorded at a sampling rate of 500 Hz using Bionomadix BN-ECG2 units and an MP160 amplifier system (Biopac Systems Inc., Goleta, CA, USA). The ECG signals were recorded using circular disposable Ag/AgCl electrodes (11 mm diameter, pregelled, 40 mm foam electrode-EL501, Biopac Systems, Goleta, CA, USA) laced using the Lead-I method. ECG signals were then processed to extract features related to cardiac activity in the following steps: (1) Acquired ECG signals are preprocessed using a band-pass filter with a pass band of 5 Hz to 15 Hz [47] to minimize the effect of muscle artifact, 60 Hz interference, baseline wander, and T-wave interference [48]. (2) R-peak was detected from the preprocessed ECG signals using the “QRS detection algorithm” [48], and the R-peak to R-peak intervals (RRIs) were calculated using the interval between R-peak to R-peak. The detected R-peaks were filtered as normal to normal (NN) intervals, and the criterion for NN intervals was defined as 600–1200 ms based on previous work [49]. Data from RRI that did not meet this criterion were excluded from the analysis. (3) The standard deviation of SDNN was obtained by calculating the standard deviation across normal RRIs [50]. (4) The pNN50 (%) was calculated from the percentage of adjacent RR intervals that differ by more than 50 ms [51]. (5) Successive RRIs were resampled at 4 Hz to convert to time series data and are then analyzed in the HRV spectrum using the fast Fourier transform (FFT, Hanning window technique). (6) The HRV spectrum was categorized into frequency bands for a VLF band ranging from 0.0033–0.04 Hz (sympathetic activity) and an HF band ranging from 0.15–0.4 Hz (parasympathetic activity); moreover, the power for each frequency band was extracted [40,52]. (7) *ln* VLF and *ln* HF (that is, normalized HRV) were calculated using the natural log (assuming *ln*) from the power spectrum values (VLF and HF powers). (8) The normalized HRV plot was categorized into nine zones (that is, Zones 1 to 8 and a reference zone) to assess the autonomic balance of the sympathetic and parasympathetic nervous systems (SNS and PNS) [40,53], as shown in Figure 2. All signal processing and data analyses were performed using MATLAB (2020b, Mathworks Inc., Natick, MA, USA).

### 2.4. Statistical Analysis

This study follows a “within-subject” design for 2D and VR viewing conditions. Moreover, ANCOVA was applied in this study because a paired *t*-test between after-viewing conditions is not able to consider the viewers before the state. ANCOVA can assess dependent variables of post-viewing states between the two conditions by considering the pre-viewing state baseline as a covariate [2,54]. Statistical significance at the 95% significance level (*p* > 0.05) was controlled by the Bonferroni correction as a conservative test to protect against Type 1 errors caused by multiple comparisons based on the number of each hypothesis (i.e., α = 0.05/n) [55,56]. For this study, the statistically significant levels of SSQ and cardiac features were set to 0.0125 (*N*, *O*, *D*, and total SSQ scores resulting in α = 0.05/4) and 0.0083 (heart rate, SDNN, pNN50, *ln* VLF, *ln* HF, and *ln* VLF/*ln* HF ratio resulting in α = 0.05/6), respectively. The effect size to verify the practical significance was analyzed using the partial eta-squared value (ηp^2^) corresponding to an F-test. The standard values for the practical significance of 0.01, 0.06, and 0.14 are generally regarded as a small, medium, and large, respectively [57]. A partial correlation was used to analyze the correlation between the SSQ scores and cardiac features (post-viewing condition), considering the pre-viewing condition as a covariate [29]. Correlation coefficients of 0.00–0.09, 0.10–0.39, 0.40–0.69, 0.70–0.89, and 0.90–1.00 represent negligible, weak, moderate (good), strong, and very strong correlations, respectively [58]. All statistical data analyses were conducted using IBM SPSS Statistics 21.0 for Windows (SPSS Inc., Chicago, IL, USA).

### 2.5. Classification

This study used four basic machine-learning algorithms (LDA, KNN, DT, and LSVM) to verify the features. Optimization results for each classification method were obtained with 10-fold cross-validation by using parameter optimization for classifiers in the classification learner, a MATLAB toolbox. The options for the optimizer are as follows: optimizer, Bayesian optimization; acquisition function, expected improvement per second plus; iterations, 100; training time limit, no; and validation, tenfold cross-validation. The accuracy, recall, precision, F1 score, and receiver operating characteristic (ROC) curve, with the area under the ROC curve (AUC) as the metric, were analyzed to determine the performance of each classification. Additionally, a permutation test was conducted to determine the confidence of the classifiers. The permutation data were repeatedly classified for 10,000 iterations for each classifier (2020b, MathWorks Inc., Natick, MA, USA).

## 3. Results

### 3.1. SSQ Scores

All SSQ scores (i.e., *N*, *O*, *D*, and total SSQ) in the VR viewing condition are higher than those in the 2D viewing condition. As seen in Figure 3, the ANCOVA analysis reveals a significant difference in post-viewing condition for *N*, *O*, *D*, and total SSQ scores with the pre-viewing condition as a covariate (*N* score: F_1,53_ = 75.948, *p* < 0.001, with a large effect size (ηp^2^ = 0.589); *O* score: F_1,53_ = 94.215, *p* < 0.001, with a large effect size (ηp^2^ = 0.640); *D* score: F_1,53_ = 91.157, *p* < 0.001, with a large effect size (ηp^2^ = 0.632); total SSQ score: F_1,53_ = 192.424, *p* < 0.001, with a large effect size (ηp^2^ = 0.784)).

### 3.2. Cardiac Activity

For the VR viewing condition, the heart rate, SDNN, pNN50, and *ln* HF are lower than those in the 2D viewing condition. Moreover, the *ln* VLF and *ln* VLF/*ln* HF ratios are larger than those under the 2D viewing conditions. As seen in Figure 4, the ANCOVA analysis reveals a significant difference in post-viewing condition for SDNN, pNN50, *ln* VLF, *ln* HF, and *ln* VLF/*ln* HF ratio with the pre-viewing condition as a covariate (SDNN: F_1,53_ = 15.244, *p* < 0.001, with a large effect size (ηp^2^ = 0.223); pNN50: F_1,53_ = 11.212, *p* < 0.0083, with a large effect size (ηp^2^ = 0.175); *ln* VLF: F_1,53_ = 37.031, *p* < 0.001, with a large effect size (ηp^2^ = 0.411); *ln* HF: F_1,53_ = 56.352, *p* < 0.001, with a large effect size (ηp^2^ = 0.515); *ln* VLF/*ln* HF ratio: F_1,53_ = 50.513, *p* < 0.001, with a large effect size (ηp^2^ = 0.488)]. Additionally, the heart rate shows no significant result [F_1,53_ = 2.456, *p* < 0.001, with a small effect size (ηp^2^ = 0.044)).

### 3.3. Normalized HRV (Autonomic Balance)

The autonomic balance was plotted in nine zones of two dimensions using *ln* VLF (*x*-axis) and *ln* HF (*y*-axis) values for each 2D and VR viewing condition. Considering the results, the 2D condition pattern is stabilized before and after viewing, and the 3D condition pattern can be observed to be significantly changed. As seen in Figure 5, the autonomic balance is mainly located within the reference zone prior to the 2D viewing condition and almost remains within that reference zone without significant changes after viewing. However, in the VR viewing condition, the autonomic balance moves consistently from within the reference zone before viewing to almost into Zone 5 after viewing. From these results, VR viewing destabilizes the autonomic balance and shifts it to a generally activating sympathetic and deactivating parasympathetic nerve.

### 3.4. Correlation Analysis

Partial correlation is present between the total SSQ score and significant cardiac features in the post-viewing condition, with residual covariates in the pre-viewing condition. Figure 6 shows the plot of residuals for the total SSQ score and cardiac features with linear regression lines. The correlation coefficients between the total SSQ scores and cardiac features (i.e., heart rate, SDNN, pNN50, *ln* VLF, *ln* HF, and *ln* VLF/*ln* HF ratio) in the post-viewing condition are significantly different (SDNN: *r* = −0.381, *p* < 0.01; pNN50: *r* = −0.377, *p* < 0.01; *ln* VLF: *r* = 0.653, *p* < 0.001; *ln* HF: *r* = −0.711, *p* < 0.001; *ln* VLF/*ln* HF ratio: *r* = 0.677, *p* < 0.001). Additionally, note that the heart rate is not significantly different (*r* = −0.160, *p* > 0.05).

### 3.5. Classification Performance and Permutation Test

In this study, the 2D and VR viewing conditions are classified. The features for the classification are only five statistically significant indicators (SDNN, pNN50, *ln* VLF, *ln* HF, and *ln* VLF/*ln* HF ratio). The heart rate is not included in the classification features. The optimized hyperparameters are as follows: the data for all methods are standardized; the number of neighbors for KNN is 28, and Euclidean distance is applied; the split criterion for DT is maximum deviance reduction, and the maximum number of splits is 2; the box constraint level for LSVM is 0.0706. The accuracies, recalls, precisions, f1 scores, and AUCs of the four classifiers (LDA, KNN, DT, and LSVM) are listed in Table 2. The ROC curves for all classifiers are shown in Figure 7. All the classifiers are significant (*p* < 0.0001) in the permutation test. Figure 8 shows the permutation test’s accuracy distributions for all the classifiers (10,000 iterations).

### 3.6. Real-Time System to Monitor MS

The developed real-time system for assessing MS comprises a Wireless ECG Amplifier (Bionomadix BN-ECG2 units), a power supply (MP160), and a personal computer for analysis. It can classify an MS or normal state in a VR environment using an HMD device. This system monitors user behavior via webcam and VR scenes and confirms the results of cardiac features in both the time and frequency domains. In addition, the results depending on time for assessing MS can be confirmed using a sliding bar for each time log. The system was developed using MATLAB App designer (2020b, Mathworks Inc., Natick, MA, USA), and signal processing is performed using the MATLAB toolbox (2020b, Mathworks Inc., Natick, MA, USA), as shown in Figure 9. A real-time system was constructed to record the onset-trigger for the start of the HMD device using user datagram protocol communication to synchronize with the computer for MATLAB. As shown in Figure 10, the proposed method for classifying the MS state involves the processes of signal measurement and synchronization, pre-processing, feature extraction, and classification.

## 4. Discussion

This paper proposes a quantitative method for measuring MS based on cardiac activity and a real-time system for assessing the same. This study confirms whether 2D and VR viewing conditions cause MS in subjects based on SSQ scores. The subjects in the VR viewing condition demonstrably experience MS but not in the 2D viewing condition, because the SSQ scores show a statistically significant difference between the 2D and VR viewing conditions. This study yields four significant findings: (1) The cardiac features, such as SDNN, pNN50, *ln* VLF, *ln* HF, and *ln* VLF/*ln* HF ratio, show a significant difference between the 2D and VR viewing conditions based on the ANCOVA analysis. (2) Considering the normalized HRV, the autonomic balance is located within the reference zone in the normal state. However, it moves almost into Zone 5 in MS. (3) The correlation coefficients between the total SSQ score and significant cardiac features are in the range of −0.377 to −0.711 (for SDNN, pNN50, and *ln* HF) or 0.653 to 0.677 (for *ln* VLF and *ln* VLF/*ln* HF ratio), indicating significant negative or positive correlations, respectively, in the range of weak to strong. (4) Among the algorithms for classifying MS, the highest average accuracy, recall, precision, F1 score, and AUC of MS classification results using the LSVM achieved 91.1, 96.4, 87.1, 91.5, and 0.96 (10-fold cross validation), respectively. The ROC curve is a useful tool when predicting the probability of a binary classification. In general, skillful models reveal curves that bow up to the top left of the plot. When the curve is close to the point (0,1), it is evaluated to be the perfect model. In addition, recall and precision are defined as the ratio of correctly classified positive samples to a total number of classified positive samples (either correctly or incorrectly) and the ratio between the numbers of positive samples correctly classified as positive to the total number of positive samples. Recall and precision are metrics used to assess the performance of classification algorithms since accuracy alone is not sufficient to understand, and improving precision typically reduces recall and vice versa. A good classification model needs to strike the right balance between precision and recall [59]. F1 score is used by combining precision and recall to obtain a balanced classification model. The proposed LSVM-based classification model was confirmed to have good performance. The permutation test generates a null distribution by calculating the accuracy of the classifier on 10,000 different permutations from the experimental data set, and the permutation-based *p*-values are used to assess the competence of a classifier. This study confirmed that the classification model for the MS reveals a significant result in the permutation test (10,000 iterations, *p* < 0.0001).

A significant difference can be observed in the cardiac activity results of individuals with subjective symptoms of MS. In this study, both the SNS’s activation and the PNS’s deactivation are determined from the normalized HRV indices in subjects that experience MS. The normalized HRV features have been demonstrated to best reflect the SNS and PNS activities [60,61]. The findings of this study are consistent with those of previous studies in the following aspects: (1) SNS activation (i.e., increasing VLF or LF power) [1,62,63,64] and (2) PNS deactivation (i.e., decreasing HF power) [1,58,61]. Physiological and morphological connections exist between the vestibular and autonomic nervous systems (i.e., the SNS and PNS) [65,66]. Conflicting inputs of afferent signals from visual, vestibular, and somatosensory signals cause MS, and MS typically transmits vestibuloautonomic responses in humans [67]. The activation of SNS can be interpreted as a defensive reaction against the sensation of nausea [68] and is closely related to physiological stress [69]. In a study combining HRV and functional magnetic resonance imaging, brain regions of the medial prefrontal cortex (MPFC) were reported to correlate significantly with the PNS activation during moderate and strong nausea. The MPFC, known to reflect automatic modulatory regions, may have an excitatory influence on the autonomic control regions. MS is closely associated with the switch from the inhibitory to excitatory influences on the cardiovagal outflow [70]. In addition, SDNN reflects both the sympathetic and parasympathetic influences, and pNN50 corresponds to the parasympathetic neural regulation from heart activity [71]. This study verifiably demonstrates that the SDNN and pNN50 show a significant difference between the MS and normal states, and the results of this study agree with those of previous studies [44,72]. Moreover, cardiac-related features are verified to be closely related to the phenomenon of MS and are significant indicators for quantitative assessment of MS.

In addition, many previous studies have reported that MS is strongly associated with the cognitive load caused by inconsistencies or conflicts among different types of sensory information (i.e., visual and motor) [2,16,73,74,75]. Sensory conflict is a well-known theory that explains MS, and it can be caused by a mismatch or inconsistency between the visual and vestibular senses during HMD usage [76]. MS can lead to distortions and delays in visual information processing by the brain due to a mismatch between sensory information; moreover, it can be interpreted as consuming excessive neural resources for processing the massive visual information in VR compared to that in 2D [2,16]. Cardiac activity is strongly associated with cognitive load. The heart transmits sensory information related to its activities to the brain based on afferent pathways, and this phenomenon is known as heartbeat-evoked potentials (HEPs) [1,77]. Cognitive processing in the brain is influenced by the changes in heart rhythm via the afferent and efferent pathways between the heart and brain [77,78,79]. Our previous work demonstrated that increasing cognitive load is correlated with an irregular pattern of heart rhythms (that is, increasing SDNN and pNN50), increasing the activation of SNS, and decreasing the activation of PNS [40]. In previous studies, significant results of cardiac activity have also been confirmed as an indicator of cognitive load [80,81,82,83]. Thus, the findings in this study related to changes in cardiac activity can suggest that the cognitive load can be interpreted as conflict among sensory inputs and processing large amounts of visual information. In addition, a quantitative method and real-time system for monitoring MS are developed in this study. Many previous studies have attempted to perform quantitative measurements using EEG, HEP, vision responses, and multi-modal fusion. These studies reported that MS can be classified with 74.7% to 99.1% accuracy [2,16,24,25,26,27,28,29,30,31,32,33,34,35]. In this study, a competitive classification accuracy (91.1%) of MS compared to previous studies was achieved using a single measurement. In addition, the approach used in previous studies has high complexity and disadvantages for measuring or evaluating MS, such as the measurements requiring sensor attachments, as well as low usability compared to cardiac activity. The proposed method also has the advantage of minimizing limitations such as complex and expensive equipment, inconvenience, and the burden of sensor attachment, which can be further minimized by replacing the implemented ECG sensor with photoplethysmography, compared to approaches using EEG or multimodal fusion, as in past studies. The method proposed in this study cannot be judged to have improved performance or contribution compared previous studies, but there will be differences in potential fields of application based on the advantages and disadvantages of each measurement method. For example, EEG or other features can be used primarily for precise measurements of MS because they have the advantage of measuring more important and significant data. We believe that the proposed method will have a contribution that can be applied to fields requiring fast and convenient measurement of MS based on usability, which is simple to measure. In addition, the symptoms of MS are known to be caused by various factors, such as user characteristics (i.e., viewing time, age, gender), viewing environment (i.e., gaze angle, fixation, field of view), device, and VR content (i.e., resolution). The relationship between the MS state and these causal factors should be verified to minimize the sensitivity of the MS. To verify each condition’s effect on the MS, the development of a methodology for quantitatively evaluating the MS state should be preceded. Thus, the proposed method can contribute to improving the VR experience for viewers by minimizing MS.

This study has several limitations. (1) This study selected an SSQ questionnaire to perform the subjective measurement of MS, because the SSQ questionnaire is the best-known test for the subjective measurement for MS. However, traditional SSQ questionnaires were developed for measuring cybersickness and may differ significantly from today’s VR experience. In future studies, recently revised questionnaires need to be considered (e.g., virtual reality sickness questionnaire [84]), including the traditional SSQ questionnaire. (2) Previous studies related to MS have often reported conflicting results. For example, after experiencing the phenomena of MS, one study reported an increase in HR [36], whereas another identified a decrease in HR [37]. This study confirmed a decrease in HR consistent with the previous study [37], but it showed no statistically significant difference between 2D and HMD. These discrepancies in study conclusions may be attributed in part to experimental and procedural differences [85]. Alternatively, they could be attributable to differences in VR content or HMD device. Thus, it cannot be determined that the proposed method can measure all of the MS symptoms caused by various conditions. This limitation needs to be validated via future studies.

## 5. Conclusions

This paper proposes a method for assessing MS caused by watching VR content on an HMD using cardiac features. This study demonstrates a significant difference in cardiac features between the MS and normal states, and the classification performance achieves an average accuracy of 91.1%. In addition, a real-time system was developed for continuously monitoring MS using significant cardiac features. The proposed method can quantify the level of MS severity and determine the optimal viewing factors of a VR environment (i.e., viewer characteristics, viewing condition, VR content factors, and HMD device factors) to minimize the symptoms of MS. This research will consequently help improve the viewing environment and lead to the establishment of viewer-friendly VR. However, since MS is strongly correlated with various factors, such as age and gender [86], further research should be conducted to increase the generality of the classifiers in subjects of different ages and genders. 

## Figures and Tables

**Figure 1 sensors-22-06213-f001:**
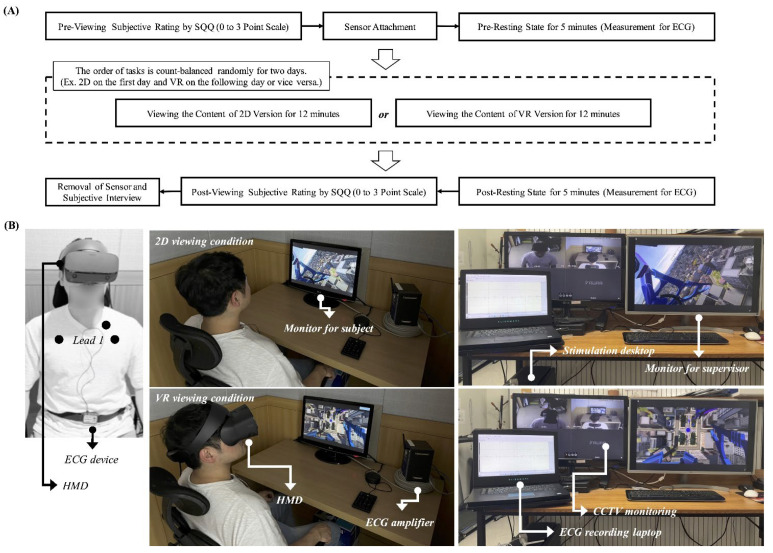
(**A**) Experimental procedure and (**B**) environment.

**Figure 2 sensors-22-06213-f002:**
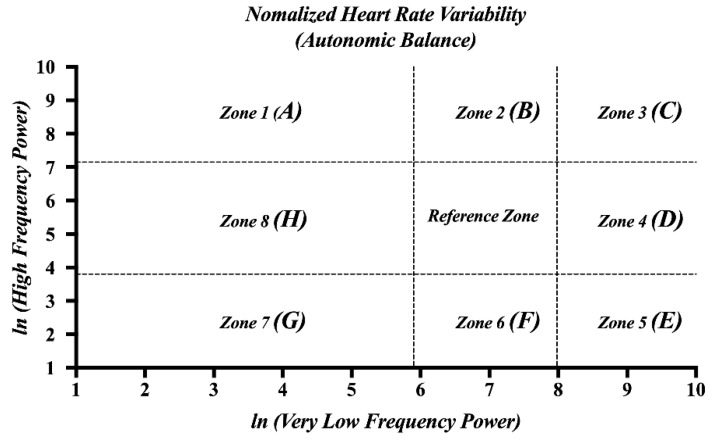
Zone definitions of normalized HRV. (*A*) Zone 1: High parasympathetic/low sympathetic. (*B*) Zone 2: High parasympathetic/normal sympathetic. (*C*) Zone 3: High dual autonomic tone. (*D*) Zone 4: High sympathetic/normal parasympathetic. (*E*) Zone 5: High sympathetic/low parasympathetic. (*F*) Zone 6: Normal sympathetic/low parasympathetic. (*G*) Zone 7: Low sympathetic and parasympathetic. (*H*) Zone 8: Low sympathetic/normal parasympathetic.

**Figure 3 sensors-22-06213-f003:**
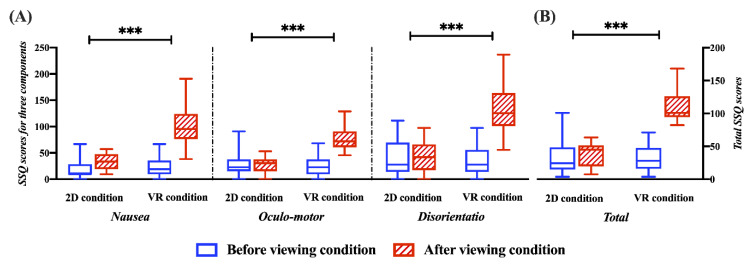
Representation of SSQ scores for MS between the 2D and VR conditions based on the ANCOVA test (*** *p* < 0.001). (**A**) SSQ items of nausea, oculomotor responses, and disorientation. (**B**) Total SSQ score.

**Figure 4 sensors-22-06213-f004:**
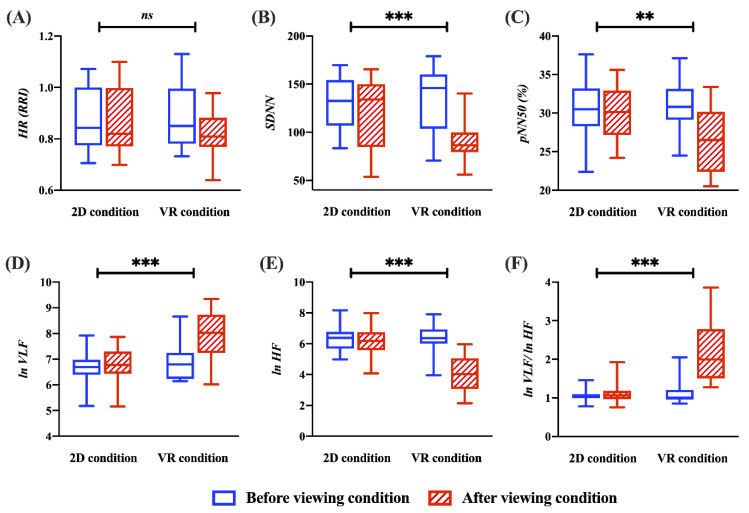
Representation of cardiac activity for MS between the 2D and VR conditions based on the ANCOVA test (*** *p* < 0.001 and ** *p* < 0.0083). (**A**) RRI (heart rate). (**B**) SDNN. (**C**) pNN50. (**D**) *ln* VLF. (**E**) *ln* HF. (**F**) *ln* VLF/*ln* HF ratio.

**Figure 5 sensors-22-06213-f005:**
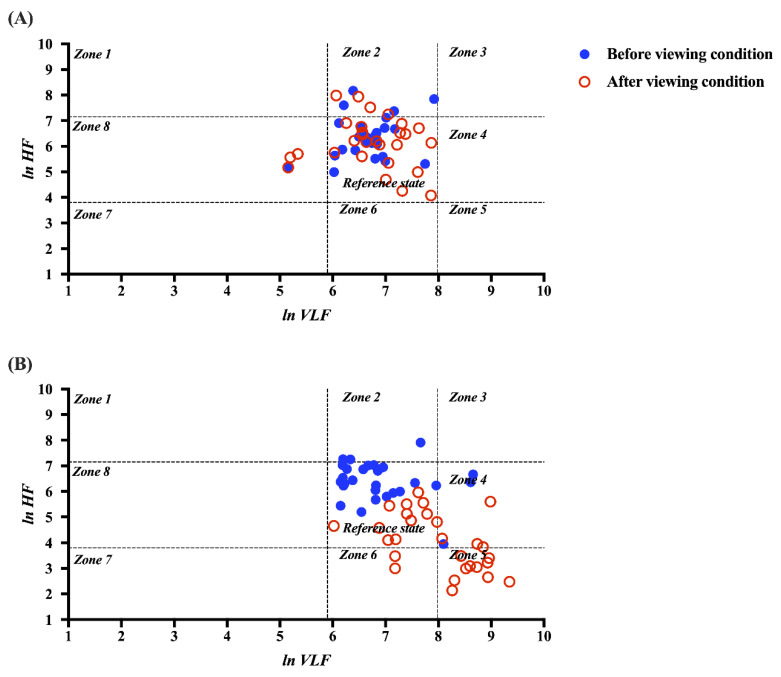
Comparison of the autonomic balance (normalized HRV) before and after the (**A**) 2D and (**B**) VR viewing conditions, structured by predefined zones of distribution.

**Figure 6 sensors-22-06213-f006:**
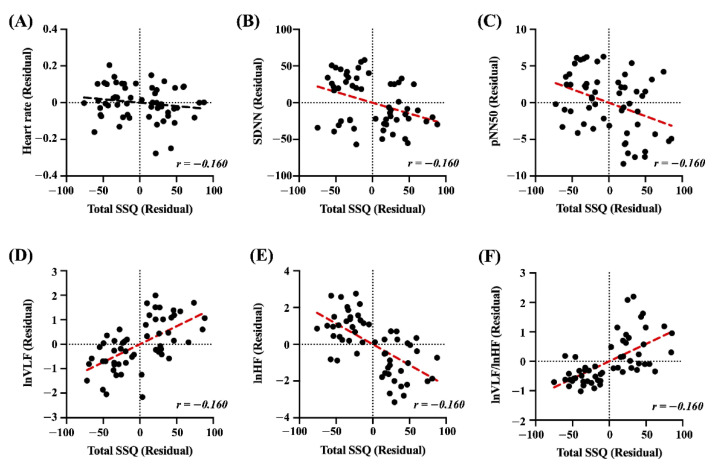
Results of the partial correlation analysis between the total SSQ score and cardiac features (red dotted line, *p* < 0.001). (**A**) Heart rate. (**B**) SDNN. (**C**) pNN50. (**D**) *ln* VLF. (**E**) *ln* HF. (**F**) *ln* VLF/*ln* HF ratio.

**Figure 7 sensors-22-06213-f007:**
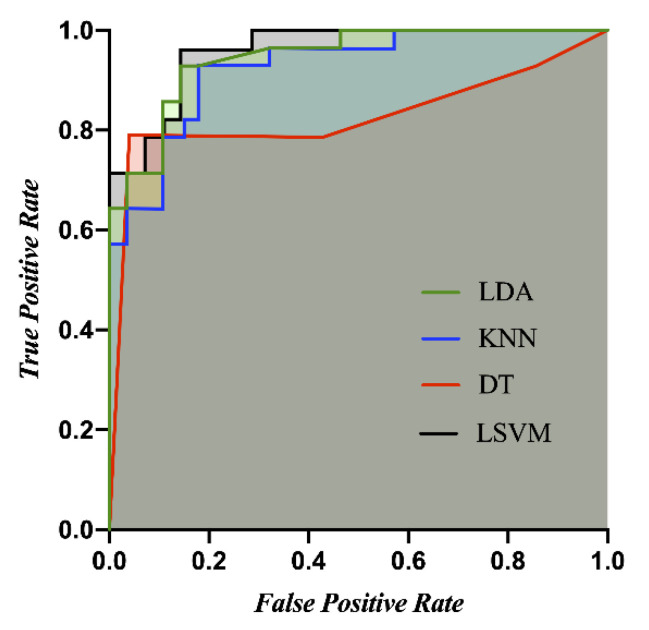
ROC curves for 10-fold cross validation according to the four classifiers (LDA, KNN, DT, and LSVM).

**Figure 8 sensors-22-06213-f008:**
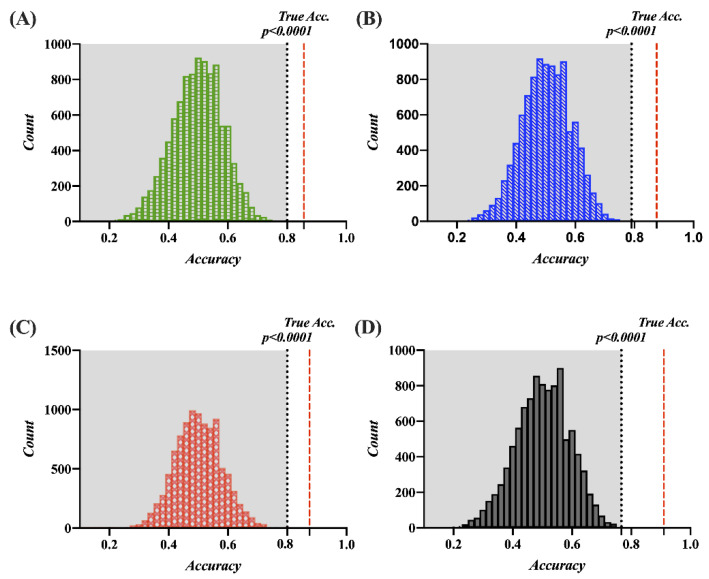
Results of the distributions for the four classifiers for the permutation test (*p* < 0.0001). (**A**) LDA. (**B**) KNN. (**C**) DT. (**D**) LSVM.

**Figure 9 sensors-22-06213-f009:**
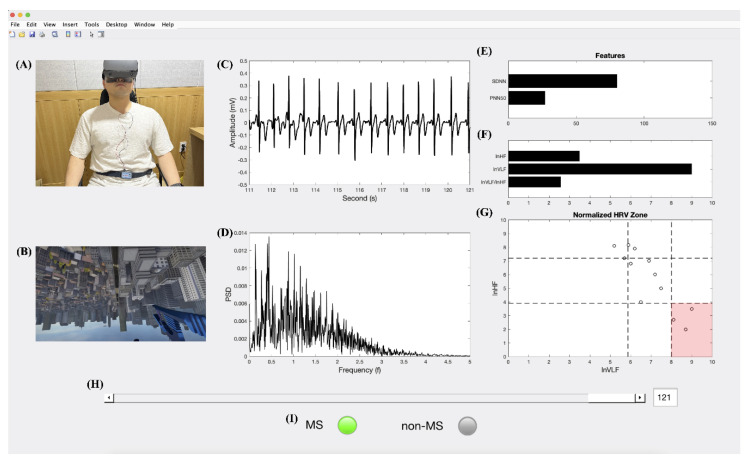
Real-time system for assessing MS using cardiac activity. (**A**) User monitor cam. (**B**) VR scene. (**C**) ECG raw signals and detecting the R-peaks. (**D**) HRV by FFT. (**E**) Results of the cardiac time domain indices (i.e., SDNN and pNN50). (**F**) Results of the cardiac frequency domain indices (i.e., *ln* HF, *ln* VLF, and *ln* VLF/*ln* HF ratio). (**G**) Results of the nine zone in autonomic balance by the normalized HRV. (**H**) Sliding bar in each time log. (**I**) The binary decision for MS.

**Figure 10 sensors-22-06213-f010:**
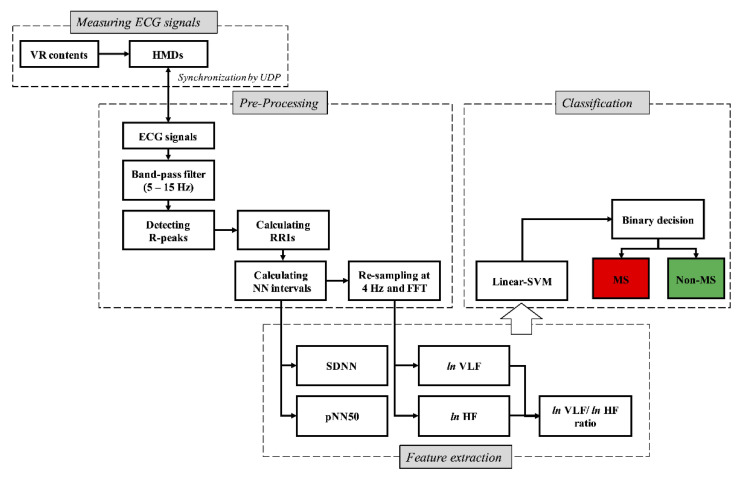
Flowchart for the proposed method of classifying motion sickness state (two-class).

**Table 1 sensors-22-06213-t001:** Summary of MS measurement literature.

Measurement		Platform	Content	Participants	Classification Performance	Paper
EEG	alpha (8–13 Hz) and theta (4–7 Hz) bands	360-degree VR-based dynamic 3D environment	driving simulation	7 subjects	95% (5-fold cross-validation)	[24]
	alpha (8–12 Hz) band	360-degree projection	driving simulation	6 subjects	86.92% (leave-one-out cross-validation)	[25]
	delta (0.1–3 Hz), theta (4–7 Hz), alpha (8–13 Hz), beta (13–20 Hz), and gamma (21–50 Hz) bands	360-degree projection	driving simulation	6 subjects	80.7% (cross-validation) 72.1% (test data set)	[26]
	alpha and beta bands	HMD	mirror edge game	9 subjects	83.8% (cross-validation)	[27]
	theta (4–8 Hz), beta (12–30 Hz), and alpha (8–12 Hz) bands	HMD	mirror edge game	9 subjects	88.9% (3-fold cross-validation)	[28]
	theta (4–8 Hz), alpha (8–12 Hz), low-beta (12–16 Hz), high-beta (16–25 Hz), and gamma (25–45 Hz) bands	360-degree video	VR video	11 subjects	99.12% (5-fold cross-validation)	[29]
	delta, theta, alpha, and beta bands (1–30 Hz)	HMD	VR scene of road	18 subjects	79.25% (10-fold cross-validation)	[30]
	parallel-feature extraction and feature attention modules	VR vehicle-driving simulator	driving simulation	8 subjects	96.7% (leave-one-out cross-validation)	[31]
HEP	first and second components	HMD	No Limits 2 Roller Coaster Simulation	48 subjects (train: 28 and test: 20)	96.4% (10-fold cross-validation) 87.5% (test data set, 20 subjects)	[2]
Vision	pupil size change	HMD	ultimate booster experience	47 subjects (train: 24 and test: 23)	90% (10-fold cross-validation) 80% (test data set, 23 subjects)	[16]
Multi-modal	EEG, ECG, RSP, EGG, and postural sway	HMD	virtual space station environment	20 subjects	95% (10-fold cross-validation)	[32]
	EEG, center of pressure, head and waist motion trajectories	projection screen	visual streaming and car driving video	20 subjects	91.1% (10-fold cross-validation)	[33]
	ECG and RSP	HMD	roller coaster video	20 subjects	96.48% (cross-validation)	[34]
	EEG, EMG, and ECG	BioVRSea (VR goggles)	rough sea scenario	28 subjects	74.7% (10-fold cross-validation)	[35]
	stomach activity, EOG, and RSP	HMD	virtual environment game	20 subjects	77.8% (cross-validation)	[36]
	ECG, EOG, RSP, and EDA	HMD	VR contents	66 subjects	82% (cross-validation)	[37]

**Table 2 sensors-22-06213-t002:** Performance of the different types of classifiers for the 2D and VR viewing conditions.

	Accuracy	Recall	Precision	F-1 Score	AUC
**LDA**	85.7	89.3	83.3	86.2	0.94
**KNN**	87.5	92.9	83.9	88.2	0.93
**DT**	87.5	78.6	95.7	86.3	0.83
**LSVM**	91.1	96.4	87.1	91.5	0.96

## Data Availability

The datasets generated and/or analyzed during the current study are not publicly available due to confidentiality agreements but are available from the corresponding author on reasonable request.

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
