# Peer review of "Effect of Visually Induced Motion Sickness from Head-Mounted Display on Cardiac Activity"

_sensors, 2022, doi:10.3390/s22166213_

Round 1

Reviewer 1 Report

The paper is well written and easy to understand. I especially appreciate that authors do not only report accuracy, but also other relevant evaluation metrics, such as F1, as well as precision and recall. Nevertheless, I would suggest discussing which one (recall or precision) is more important for cybersickness classification?

Furthermore, the contribution of this paper is not clear at the beginning of the paper. As the authors themselves mention, many researchers utilize machine learning approaches to classify or detect CS (see Table 1). Here, some relevant papers are missing, such as [2] and [3]. More importantly, the authors should clarify their contribution already in the introduction section. I think, the life monitor of MS (first mentioned in Section 3.6.) is one of the novel features if it could be extended in such a way, that it could predict or detect CS in real-time. In the Figure, the authors also describe that the tool consists of a binary decision for MS; however, the description is too sparse.

Furthermore, authors should compare and discuss their results with related work. In related work, there are contradictory results. Sometimes cybersickness is identified when the cardiac features decrease, sometimes when they increase. For example, Dennison et al. [2] identified an increase in HR whereas Garcia-Agundez et al. [3] identified a decrease in HR. How are these results comparable with yours?

Major issues:

-       Although SSQ is still one of the most frequently used questionnaires to assess cybersickness, recent studies reported some issues and other questionnaires were proposed, e.g., VRSQ [1]. As SSQ was initially designed to evaluate simulator sickness, significantly different from today’s VR experiences, the authors should discuss why this and not another questionnaire is used for their study.

-       In Section 2.3., the authors mention that they used intervals between R peaks; however, later they mention that pNN50 was calculated. As the name already suggests, pNN50 uses intervals between normal-to-normal (NN) intervals. How did you filter NN peaks?

Minor issues:

-       The caption of Figure 2 needs to be corrected (to times (B))

-       I am not sure if “Oculus Rift OC DK2” is the correct version. The DK2 was released back in 2014 and looks different from the HMD visible in Figure 2.

[1] Kim, H. K., Park, J., Choi, Y., & Choe, M. (2018). Virtual reality sickness questionnaire (VRSQ): Motion sickness measurement index in a virtual reality environment. Applied ergonomics, 69, 66-73.

[2] Dennison, M. S., Wisti, A. Z., & D’Zmura, M. (2016). Use of physiological signals to predict cybersickness. Displays, 44, 42-52.

[3] Garcia-Agundez, A., Reuter, C., Becker, H., Konrad, R., Caserman, P., Miede, A., & Göbel, S. (2019). Development of a classifier to determine factors causing cybersickness in virtual reality environments. Games for health journal, 8(6), 439-444.

Author Response

Dear Reviewer,

Thank you for reading and reviewing our manuscript. This will help us improve its scientific caliber. We did our best to address all your concerns and revise our manuscript, as you suggested. Given below are detailed responses to the attached file. We would like to express our gratitude once more.

Sincerely,

Reviewer 2 Report

The authors propose a method for assessing motion sickness caused by watching VR content. In general, the paper is well written, and it can be the basis for further experiments. Nevertheless, there are some issues that the authors have to address:

1. Please include a discussion about the results in comparison with the information of Table 1

2. Please include a discussion about the main disadvantages of works shown in table 1 (what is the research gap?)

3. Is Figure 1 free of author’s copyright?

4. Please include a graphic to describe the proposed method. 

5. Please include a discussion of Figures 8 and 9. 

6. In Section 3.6 the authors mention an ECG Bluetooth sensor. Is it the same sensor as mentioned in section 2.3? because figures 2 and 10 show the same ECG device. 

7. Line 136, section 2.3. is it correct Ag-Cl electrodes? (or Ag/AgCl electrodes).

8. Please include a description of the limitations of this study

9. Line 94, section 2.2. Please include the criteria to select the VR content (in this case why did you decide to use “Ultimate Booster Experience”?, Is this a gold standard?).

10. The authors state that (lines 44-45) “studies on minimizing MS are necessary, which can contribute to improving the VR experience”. In the discussion section, please include a description of the following:

a. In this line what are the implications of your results? How can the proposed method help minimize MS?

b. How to consider that the results are not biased due to the VR content, or the head-mounted display used?

11. The authors state that “none of the participants had previously used HMD to view VR content”. Then, how can you ensure that the results are not due to the user experience? in other words, the MS is not due to a first use effect of HMD. What about participants who often use HMD?

12. Figure 10. Please include a description of how you synchronized the VR view with the ECG data (and results).

Author Response

(The authors gave the same response as above.)

Reviewer 3 Report

Authors in this work assess the influence of the Induced Motion Sickness while watching a VR content in a Head mounted display on the human Cardiac Activity. The domain of research is very important and much progress have been done in this direction in the recent years. Authors successfully designed the subjected test environment. Number of subjects involved in the test is sufficient for concluding the observations. Results have been well-depicted at the end of the paper with proper analysis. However, there are a few suggestions which should be incorporated in the manuscript. The comments are as follow;

1. In the Introduction section it is mentioned that a major drawback of the existing approaches is that they employ a huge burden of the attached sensors and a huge cost. In my opinion a high cost is a surely a limitation. But if we consider EEG or other tracking sensors although being costly they capture more important and significant data. Therefore, a comparison on basis of Cost between two different devices used for different tasks is not justified. Authors are suggested not to mention this as a limitation or contribution of the current article. 

2. Authors have not presented a description of how the Sensor device is calibrated and attached with the subjects. How many sensors were used and whether they were separate for every subjects? Details of such queries are missing in the current article.

3. A certain type of information could be obtained using the attached ECG. How that kind of data is different from the type of data captured using other sensors (mentioned in Table 1) should be described. And Authors should also justify how the data received using ECG can help in estimating motion sickness in comparison to the sensors used and presented in Table 1. Just a mere usage of ECG only because no literature have done it before does not qualify to be a contribution. Authors should justify the same.

4. Usually a 5 point scale is used for the single stimulus subjective tests as recommended in ITU-T Rec. P913. The rationale behind a 4 point scale starting from 0 should be mentioned in the paper.

5. In Figure 2(A), it is mentioned that there is an option between 2D and VR for viewing the content. Was it optional or they did both and answered the SSQ? A clarification is needed in the manuscript.

6. Typographical errors such as in Page 6 Figure 2, Figure 4, etc. should be addressed in the revised version. 

Author Response

(The authors gave the same response as above.)

Round 2

Reviewer 1 Report

The authors answered to almost all my questions, apart from the first one (“… I would suggest discussing which one (recall or precision) is more important for cybersickness classification?“). I would still suggest to shortly discuss which one is more important for your purposes, also in order to justify the importance of reporting these values.

Author Response

Dear Reviewer,

Thank you for reading and reviewing our manuscript, and helped us improve it to a better scientific level. We did our best to address all your concerns and revise our manuscript, as you suggested. Detailed answer to the suggestion is shown in the attached file. Once again, we would like to express our gratitude.

Sincerely,

Reviewer 2 Report

The authors have addressed all issues/suggestions of the previous review.

Author Response

Comments and Suggestions for Authors

The authors have addressed all issues/suggestions of the previous review.

Dear Reviewer,

Thank you for reading and reviewing our manuscript, and helped us improve it to a better scientific level. We are pleased that our manuscript has been judged suitable for publication in the Sensors. Once again, we would like to express our gratitude.

Sincerely,

Reviewer 3 Report

Authors have carefully addressed all my concerns. I do not have further queries from the revised manuscipt.

Author Response

Comments and Suggestions for Authors

Authors have carefully addressed all my concerns. I do not have further queries from the revised manuscript.

Dear Reviewer,

Thank you for reading and reviewing our manuscript, and helped us improve it to a better scientific level. We are pleased that our manuscript has been judged suitable for publication in the Sensors. Once again, we would like to express our gratitude.

Sincerely,